

# Genetic alteration of histone lysine methyltransferases and their significance in renal cell carcinoma

Libin Yan[1,2], Yangjun Zhang[1,2], Beichen Ding[1,2], Hui Zhou[1,2], Weimin Yao[1,2] and Hua Xu[1,2]

[1] Urology, Tongji Hospital,Tongji Medical College, Huazhong University of Science Technology, Wuhan, Hubei, China
[2] Institute of Urology of Hubei Province, Wuhan, China

## ABSTRACT

**Background:** Histone lysine methyltransferases (HMTs), a category of enzymes, play essential roles in regulating transcription, cellular differentiation, and chromatin construction. The genomic landscape and clinical significance of HMTs in renal cell carcinoma (RCC) remain uncovered.

**Methods:** We conducted an integrative analysis of 50 HMTs in RCC and discovered the internal relations among copy number alterations (CNAs), expressive abundance, mutations, and clinical outcome.

**Results:** We confirmed 12 HMTs with the highest frequency of genetic alterations, including seven HMTs with high-level amplification, two HMTs with somatic mutation, and three HMTs with putative homozygous deletion. Patterns of copy number and expression varied among different subtypes of RCC, including clear cell renal cell carcinoma, papillary cell carcinoma, and chromophobe renal carcinoma. Kaplan–Meier survival analysis and multivariate analysis identified that CNA or mRNA expression in some HMTs were significantly associated with shorter overall patient survival. Systematic analysis identified six HMTs (ASH1L, PRDM6, NSD1, EZH2, WHSC1L1, SETD2) which were dysregulated by genetic alterations as candidate therapeutic targets.

**Discussion:** In summary, our findings strongly evidenced that genetic alteration of HMTs may play an important role in generation and development of RCC, which lays a solid foundation for the mechanism for further research in the future.

Corresponding author
Hua Xu, xuhuawhu@163.com

# INTRODUCTION

Renal cell carcinoma (RCC) accounts for nearly 5% of adult malignancies with about 63,920 new cases and 13,860 deaths in the United States (*Siegel et al., 2014*). RCC is histologically classified into several subtypes, including clear cell renal cell carcinoma (ccRCC), papillary cell carcinoma (pRCC), and chromophobe renal carcinoma (chRCC), among which ccRCC is the most common subtype and accounts for approximately 70–80% of all RCC (*Yan, Mackinnon & Al-Ahmadie, 2009*). In addition to surgical treatment, current targeted therapies have slightly improved overall survival in patients

with advanced disease. Histone lysine methyltransferase (HMT) and demethylases control the process of histone lysine methylation, which is a crucial part of epigenetics (*Greer & Shi, 2012*; *Albert & Helin, 2010*). To date, over 50 human HMTs have been revealed (*Herz, Garruss & Shilatifard, 2013*). In the structure, the HMTs are a group of various proteins, which are typically categorized by two functional enzyme families, the SET-domain-containing methyltransferase and DOT1L lysine methyltransferase (*Albert & Helin, 2010*; *Herz, Garruss & Shilatifard, 2013*) (Table 1). Recent researches have shown that HMT dysregulation leads to uncontrollable histone methylation pathways and contributes to the pathogenesis of many human cancers, including RCC (*Niu et al., 2012*; *Roy, Walsh & Chan, 2014*; *Shen & Laird, 2013*; *Vieira-Coimbra, Henrique & Jeronimo, 2015*; *Pires-Luís et al., 2015*). Several studies indicated that the methyltransferase gene SETD2 was frequently mutated during epigenetic progress in RCC (*Tiedemann et al., 2016*). It was also demonstrated that SETD2-mutated patients were characterized by loss of function of nucleosome structure, replisome occupancy, replication fork progression, and DNA repair by homologous replication (*Kanu et al., 2015*). Also, EZH2, a histone 3 lysine 27 methyltransferase, proved to be associated with poor prognoses in ccRCC (*Wagener et al., 2008*, *2010*). Increasing evidences showed that genetic alterations of several HMTs play crucial roles in oncogenesis (*Shen & Laird, 2013*; *Roy, Walsh & Chan, 2014*; *Tian et al., 2013*). By far, there is no systematic analysis of genomic aberration of HMTs in RCC. Furthermore, the clinical relevance of genetic alterations in each HMT in RCC remains unclear. Our goal, therefore, is to demonstrate the genomic alteration of HMTs in RCC and assess their diagnostic and prognostic potential.

# MATERIALS AND METHODS

## Samples with genomic and clinical data

The DNA copy number, gene expression, mutation, clinicopathological data and overall survival datasets of 882 RCC samples used in this research were obtained from The Cancer Genome Atlas (TCGA) at https://genome-cancer.ucsc.edu, including 528 ccRCC, 288 pRCC, and 66 chRCC samples. A total of 50 human HMTs were analyzed. In addition, 78 samples of Tokyo university downloaded from cBioportal were added to persuasively compare the tumor stage and overall survival between 62 SET-domain mutated patients and 16 non-SET-domain mutated patients. The copy number of HMTs is generated by the copy number algorithm genomic identification of significant targets in cancer algorithm analysis: "−2" represents a homozygous deletion, "−1" indicates a heterozygous deletion, "0" represents diploid, "1" indicates a low-level gain, and "2" signifies a high-level amplification. For mRNA expression data, the relative expression and gene expression profiles in the RCC samples were analyzed.

## Statistical analysis

Statistical analyses were performed using the R software (*R Core Team, 2013*), Graphpad Prism (version 7.01; GraphPad, La Jolla, CA, USA), and SPSS (version 18.0; SPSS, Chicago, IL, USA). The correlations between copy number alteration (CNA) and mutation

**Table 1 Summary of identified human HMTs and their substrates.**

| Official symbol | Other aliases | Gene ID | Gene location | Histone substrates |
|---|---|---|---|---|
| SUV420H1 | CGI85; KMT5B | 51111 | 11q13.2 | H4K20 |
| SUV420H2 | KMT5C | 84787 | 19q13.42 | H4K20 |
| ASH1L | ASH1; KMT2H; ASH1L1 | 55870 | 1q22 | H3K4; H3K36 |
| SMYD1 | KMT3D | 150572 | 2p11.2 | H3K4 |
| SMYD3 | KMT3E; ZMYND1 | 64754 | 1q44 | H3K4 |
| SMYD2 | KMT3C; ZMYND14 | 56950 | 1q32.3 | H3K4; H3K36 |
| SMYD4 | ZMYND21 | 114826 | 17p13.3 | |
| SMYD5 | RRG1; RAI15 | 10322 | 2p13.2 | |
| NSD1 | KMT3B | 64324 | 5q35.2 | H3K36 |
| WHSC1 | WHS; NSD2 | 7468 | 4p16.3 | H3K36; H4K20 |
| WHSC1L1 | NSD3; pp14328 | 54904 | 8p11.23 | H3K4; H3K27 |
| DOT1L | DOT1; KMT4 | 84444 | 19p13.3 | H3K79 |
| EZH1 | KMT6B | 2145 | 17q21.2 | H3K27 |
| EZH2 | EZH1; KMT6 | 2146 | 7q36.1 | H3K27 |
| SETD3 | C14orf154 | 84193 | 14q32.2 | |
| SETD4 | C21orf18; C21orf27 | 54093 | 21q22.12 | |
| SETD6 | | 79918 | 16q21 | |
| SETD7 | KMT7; SET7; SET9 | 80854 | 4q31.1 | H3K4 |
| KMT2B | MLL2; MLL4 | 9757 | 19q13.12 | H3K4 |
| KMT2C | HAIR; MLL3 | 58508 | 7q36.1 | H3K4 |
| KMT2A | HRX; MLL; MLL1 | 4297 | 11q23.3 | H3K4 |
| KMT2D | MLL2; MLL4 | 8085 | 12q13.12 | H3K4 |
| SETD8 | SET8; KMT5A; SETD7 | 387893 | 12q24.31 | H4K20 |
| MECOM | EVI1; PRDM3 | 2122 | 3q26.2 | H3K9me1 |
| PRDM16 | MEL1; LVNC8; PFM13 | 63976 | 1p36.32 | H3K9me1 |
| PRDM13 | PFM10 | 59336 | 6q16.2 | |
| PRDM8 | PFM5 | 56978 | 4q21.21 | H3k9 |
| PRDM1 | BLIMP1 | 639 | 6q21 | |
| PRDM2 | RIZ; KMT8 | 7799 | 1p36.21 | H3k9 |
| PRDM10 | PFM7 | 56980 | 11q24.3 | |
| PRDM12 | PFM9 | 59335 | 9q34.12 | |
| PRDM6 | | 93166 | 5q23.2 | |
| PRDM14 | PFM11 | 63978 | 8q13.3 | |
| PRDM4 | PFM1 | 11108 | 12q23.3 | |
| PRDM15 | PFM15; ZNF298 | 63977 | 21q22.3 | |
| PRDM5 | BCS2; PFM2 | 11107 | 4q27 | |
| PRDM7 | PFM4; ZNF910 | 11105 | 16q24.3 | |
| PRDM9 | PFM6; MSBP3; PRMD9 | 56979 | 5p14.2 | H3K4 |
| PRDM11 | PFM8 | 56981 | 11p11.2 | |
| EHMT1 | GLP; KMT1D | 79813 | 9q34.3 | H3K9; H1.2K187 |
| EHMT2 | G9A; NG36; KMT1C | 10919 | 6p21.31 | H3K9; H3K27 |

(Continued)

| Official symbol | Other aliases | Gene ID | Gene location | Histone substrates |
|---|---|---|---|---|
| SUV39H1 | MG44; KMT1A | 6839 | Xp11.23 | H3K9 |
| SUV39H2 | KMT1B | 79723 | 10p13 | H3K9 |
| SETD1A | Set1; KMT2F; Set1A | 9739 | 16p11.2 | H3K4 |
| SETD1B | KMT2G; Set1B | 23067 | 12q24.31 | H3K4 |
| SETDB1 | ESET; KG1T; KMT1E | 9869 | 1q21.3 | H3K9 |
| SETDB2 | CLLD8; CLLL8; KMT1F | 83852 | 13q14.2 | H3K9 |
| SETD2 | SET2; HIF-1; HIP-1 | 29072 | 3p21.31 | H3K36 |
| KMT2E | MLL5 | 55904 | 7q22.3 | H3K4 |
| SETD5 | | 55209 | 3p25.3 | |
| SETMAR | Mar1; HsMar1 | 6419 | 3p26.1 | H3K36 |

status of 50 HMTs and 882 phenotypes of specimens were analyzed using Chi-square test. The CNA and mRNA expression of 46 HMTs from 882 sequenced RCC specimens were analyzed using Spearman, Kendall, and Pearson correlation tests. KMT2A, KMT2C, KMT2D, and KMT2E were excluded for lack of data. Heatmap of HMTs expression profiles in different subtypes of RCC was conducted by R statistical software. The Student's $t$-test was used in calculating differences in mRNA expression levels of each HMT between ccRCC and other subtypes. Kaplan–Meier survival curve was conducted to analysis the impact of CNA or gene expression of different HMTs on survival. Multivariate survival analysis was performed to investigate prognosis factors by Cox regression. The mRNA relative expression levels were analyzed by one-way analysis of variance. $P < 0.05$ was considered statistically significant.

## Gene set enrichment analysis

Gene set enrichment analysis (GSEA) was performed by GSEA software (Version 2.2.2), which was downloaded from the Broad Institute (http://www.broad.mit.edu/gsea). Enrichment map was generated for visualization of the GSEA results. False discovery rate value, normalized enrichment score, and adjusted $P$-value were calculated to identify the Hallmarks enriched in each phenotype.

## Cell culture

293T, HK-2 and ACHN cells were maintained in Dulbecco's modified Eagle's medium supplemented with 10% fetal bovine serum in an atmosphere at 37 °C with 5% $CO_2$. 786-O and OSRC-2 cells were cultured in RPMI-1640 supplemented with 10% fetal bovine serum in an atmosphere at 37 °C with 5% $CO_2$.

## Quantitative real-time PCR

Total RNAs were extracted by MagZol (Invitrogen, Carlsbad, CA, USA) and cDNAs were synthetized by using SYBR Premix Ex TaqTM (TaKaRa, Kusatsu, Japan). Real-time PCR was performed by SYBR Green Realtime PCR Master Mix (TOYOBO, Osaka, Japan). Amplification conditions were as follows: 95 °C for 15 s, 60 °C for 30 s, 72 °C for 30 s for

40 cycles in a 20 µl reaction mix containing 2× SYBR Green. Primers for the reaction are provided in Table S1.

## Plasmids

The shRNA plasmids for SETD2 and EZH2 knockdown were constructed from pSicoR (#11579; Addgene, Watertown, MA, USA) with target sequences of shSETD2: TAGTACACCAAGACTCCAG, and shEZH2: CCAACACAAGTCATCCCATTA. All plasmids were verified by sequencing.

## Cell viability, cell proliferation, cell migration, and invasion assays

Cell viability was assessed at 0, 24, 48, 72, and 96 h upon treatments by the 3-(4,5-dimethylthiazol-2-yl)-5-(3-carboxymethoxyphenyl)-2-(4-sulfophenyl)-2H-tetrazolium, inner salt (MTS) method (Promega, Madison, WI, USA, #0000253755) according to the manufacturer's instructions. The MTS have six replications. Cell proliferation was estimated using the cell-light EdU Apollo 568 in vitro kit (Ribobio, GuangZhou, China. #C10310-1) according to the manufacturer's instructions. Migration and invasion assays were performed using uncoated and Matrigel-coated Transwell inserts according to manufacturer's instructions. All experiments were performed in triplicate.

# RESULTS

## Genetic alterations of HMTs in renal cell carcinoma

Copy number alteration and somatic mutations are crucial mechanisms for oncogenesis or inactivating tumor suppressor genes in the occurrence and development of cancer (*Albertson et al., 2003*). We hypothesized that genetic alterations of HMTs play significant roles in the development and progression of RCC (*Pires-Luís et al., 2015*). To systematically identify HMTs' potential of being biomarkers of diagnosis and prognosis of RCC, we first analyzed copy numbers and mutations of 882 RCC samples from the TCGA database via Cancer Browser (*Yao et al., 2016*). In Cancer Browser CNA datasets of RCC, as previously described, CNA was counted as "−2," "−1," "0," "1," and "2," and the average CNA rate of 50 HMTs was −0.0066. We found a different pattern of altered copy number and mutation of HMTs in RCC. Strikingly, as showed in Table 2, seven HMTs (NSD1, PRDM6, MECOM, KMT2C, EZH2, PRDM14, and KMT2E) exhibited high-level amplification in more than 0.5% of RCC samples, and two of these seven HMTs (NSD1 and PRDM6) had a much higher amplification rate in over 10% of ccRCC samples. Three HMT genes, SETD2, SETD5, and SETMAR, showed homozygous deletion in more than 10% ccRCC samples. Intriguingly, NSD1 and PRDM6, the highest frequency of the top two HMTs with high-level amplification, were located in chromosome 5q; whereas SETD2, SETD5, and SETMAR, the highest frequency of the top three HMTs with homozygous deletion, were located in chromosome 3p. Additionally, KMT2C and SETD2 exhibited somatic mutations in more than 6% of ccRCC samples. Several studies revealed that SETD2 was frequently mutated in ccRCC and SETD2-mutated signal pathway played a vital role in the process of oncogenesis (*Li et al.,*

**Table 2** Frequency of HMT copy number alterations and mutations (%).

| Gene | Gene location | Amp | Gain | Diploid | Hetloss | Homdel | Mutation |
|---|---|---|---|---|---|---|---|
| NSD1 | 5q35.2 | 16.67 | 46.21 | 35.98 | 1.14 | 0.00 | 2.82 |
| PRDM6 | 5q23.2 | 13.45 | 44.13 | 41.10 | 1.33 | 0.00 | 0.00 |
| MECOM | 3q26.2 | 1.70 | 14.58 | 71.59 | 12.12 | 0.00 | 0.94 |
| KMT2C | 7q36.1 | 1.14 | 32.20 | 65.53 | 1.14 | 0.00 | 6.10 |
| EZH2 | 7q36.1 | 0.95 | 32.20 | 65.72 | 1.14 | 0.00 | 0.47 |
| PRDM14 | 8q13.3 | 0.57 | 12.12 | 74.43 | 12.88 | 0.00 | 0.47 |
| KMT2E | 7q22.3 | 0.57 | 32.58 | 66.67 | 0.19 | 0.00 | 1.41 |
| SETDB1 | 1q21.3 | 0.38 | 11.55 | 81.82 | 6.25 | 0.00 | 0.94 |
| ASH1L | 1q22 | 0.38 | 11.74 | 81.44 | 6.44 | 0.00 | 0.94 |
| SETD1A | 16p11.2 | 0.38 | 20.08 | 77.27 | 2.27 | 0.00 | 0.47 |
| KMT2A | 11q23.3 | 0.38 | 5.87 | 89.02 | 4.73 | 0.00 | 1.88 |
| PRDM11 | 11p11.2 | 0.19 | 6.26 | 90.51 | 3.04 | 0.00 | 0.47 |
| SMYD3 | 1q44 | 0.19 | 10.80 | 81.06 | 7.95 | 0.00 | 0.00 |
| WHSC1L1 | 8p11.23 | 0.19 | 5.68 | 68.75 | 24.62 | 0.76 | 2.82 |
| SMYD2 | 1q32.3 | 0.19 | 10.98 | 81.25 | 7.58 | 0.00 | 0.47 |
| PRDM1 | 6q21 | 0.19 | 1.52 | 71.59 | 26.52 | 0.19 | 0.47 |
| SUV39H2 | 10p13 | 0.19 | 3.22 | 84.09 | 12.31 | 0.19 | 0.00 |
| PRDM9 | 5p14.2 | 0.19 | 32.01 | 66.86 | 0.95 | 0.00 | 0.47 |
| EZH1 | 17q21.2 | 0.19 | 5.68 | 88.45 | 5.68 | 0.00 | 0.47 |
| SUV39H1 | Xp11.23 | 0.19 | 5.49 | 85.42 | 8.52 | 0.38 | 0.00 |
| PRDM7 | 16q24.3 | 0.19 | 18.56 | 77.08 | 4.17 | 0.00 | 0.47 |
| EHMT1 | 9q34.3 | 0.19 | 2.84 | 67.61 | 28.98 | 0.38 | 0.94 |
| PRDM10 | 11q24.3 | 0.19 | 5.49 | 88.83 | 4.92 | 0.57 | 0.00 |
| PRDM5 | 4q27 | 0.19 | 2.08 | 83.33 | 14.20 | 0.19 | 0.47 |
| DOT1L | 19p13.3 | 0.19 | 8.71 | 87.50 | 3.22 | 0.38 | 0.94 |
| SETD7 | 4q31.1 | 0.19 | 2.65 | 82.95 | 14.02 | 0.19 | 0.00 |
| SMYD4 | 17p13.3 | 0.19 | 5.11 | 85.61 | 9.09 | 0.00 | 0.94 |
| SETDB2 | 13q14.2 | 0.19 | 3.98 | 80.49 | 15.15 | 0.19 | 0.00 |
| SUV420H1 | 11q13.2 | 0.00 | 6.25 | 90.15 | 3.60 | 0.00 | 1.41 |
| SUV420H2 | 11q13.2 | 0.00 | 10.80 | 88.07 | 0.95 | 0.19 | 0.47 |
| SETMAR | 3p26.1 | 0.00 | 1.33 | 12.31 | 76.33 | 10.04 | 0.47 |
| SETD5 | 3p25.3 | 0.00 | 1.33 | 11.17 | 77.08 | 10.42 | 0.94 |
| EHMT2 | 6p21.31 | 0.00 | 1.89 | 79.17 | 18.94 | 0.00 | 1.41 |
| PRDM13 | 6q16.2 | 0.00 | 1.52 | 71.78 | 26.52 | 0.19 | 0.94 |
| PRDM15 | 21q22.3 | 0.00 | 9.66 | 79.55 | 10.42 | 0.38 | 0.47 |
| PRDM16 | 1p36.32 | 0.00 | 1.70 | 79.36 | 18.94 | 0.00 | 0.47 |
| SETD4 | 21q22.12 | 0.00 | 9.66 | 80.11 | 10.23 | 0.00 | 0.00 |
| SETD8 | 12q24.31 | 0.00 | 22.92 | 75.95 | 1.14 | 0.00 | 0.47 |
| PRDM12 | 9q34.12 | 0.00 | 2.46 | 68.37 | 28.98 | 0.19 | 0.00 |
| PRDM8 | 4q21.21 | 0.00 | 2.46 | 85.04 | 12.50 | 0.00 | 0.00 |
| SETD1B | 12q24.31 | 0.00 | 22.92 | 75.95 | 1.14 | 0.00 | 0.00 |
| SETD3 | 14q32.2 | 0.00 | 2.84 | 53.60 | 43.18 | 0.38 | 1.41 |

| Gene | Gene location | Amp | Gain | Diploid | Hetloss | Homdel | Mutation |
|------|---------------|-----|------|---------|---------|--------|----------|
| WHSC1 | 4p16.3 | 0.00 | 2.84 | 83.90 | 13.26 | 0.00 | 1.88 |
| KMT2D | 12q13.12 | 0.00 | 22.73 | 76.33 | 0.95 | 0.00 | 1.41 |
| PRDM4 | 12q23.3 | 0.00 | 22.92 | 75.95 | 1.14 | 0.00 | 0.00 |
| SETD6 | 16q21 | 0.00 | 18.56 | 77.27 | 4.17 | 0.00 | 0.00 |
| SMYD5 | 2p13.2 | 0.00 | 14.39 | 82.95 | 2.65 | 0.00 | 0.00 |
| PRDM2 | 1p36.21 | 0.00 | 1.33 | 79.92 | 18.75 | 0.00 | 0.47 |
| SMYD1 | 2p11.2 | 0.00 | 14.39 | 83.14 | 2.46 | 0.00 | 0.00 |
| SETD2 | 3p21.31 | 0.00 | 1.14 | 10.80 | 77.27 | 10.80 | 11.51 |

**Notes:**
Amp = high-level amplification; Gain = low-level gain; Hetless = heterozygous deletion; Homdel = homozygous deletion. Genes were ranked based on the frequency of high-level amplification.

2016a; *Wang et al., 2015*; *Li et al., 2016b*). Analogously, in our study SETD2 showed the highest mutation rate (11.51%) among 50 HMTs in ccRCC (Table 2).

Furthermore, HMTs showed different frequencies of CNA and mutation in different subtypes of RCC. Among the seven most frequently amplified HMTs, the frequencies of NSD1 and PRDM6 of high-level amplification were markedly higher in 528 ccRCC with more than 13.45% of tumor samples exhibiting amplificated when compared with that of 288 pRCC and 66 chRCC subtypes, which was less than 1.04% of tumor samples (Fig. 1A). Also, SETD2 and SETD5 exhibited the highest frequency of homozygous deletion in ccRCC. However, neither of them exhibited homozygous deletion in pRCC and chRCC subtypes (Fig. 1B). Additionally, in 213 ccRCC, 113 pRCC, and 65 chRCC, of the most commonly mutated HMTs, SETD2, and KMT2C were most frequently mutated in ccRCC and pRCC subtypes, whereas SETD2 was mutated in less than 1.54% of tumor samples and KMT2C did not exhibit mutation in chRCC (Fig. 1C). The above data indicates that the subtype of ccRCC has a higher CNA and somatic mutation frequency in several HMTs, including amplification of NSD1 and PRDM6, homozygous deletion of SETD2, SETD5, and SETMAR, and mutation of KMT2C and SETD2.

## Gene expression and CNA profiling of HMTs in renal cell carcinoma

The correlation between gene expression and copy number has been associated with the detection of human oncogenes. Therefore, we analyzed the correlation between copy number and gene expression level of 46 HMTs from 882 sequenced RCC samples which were divided into ccRCC and non-ccRCC (KIRP + KICH) groups. Four HMTs (KMT2A, KMT2C, KMT2D, and KMT2E) were excluded for lack of data. The rank correlation coefficients in the three statistical tests were similar for HMTs (Table 3). As shown in Table 3, HMTs were ranked based on the Spearman correlation coefficient, and the correlations between CNA and mRNA expression of 46 HMT genes were positive, and three of them (NSD1, WHSC1L1, SETDB1) showed a Spearman correlation coefficient greater than 0.5. ($P < 0.0001$). WHSC1L1 exhibited the highest correlation coefficient by Pearson ($r = 0.669$), Spearman ($r = 0.689$) and Kendall

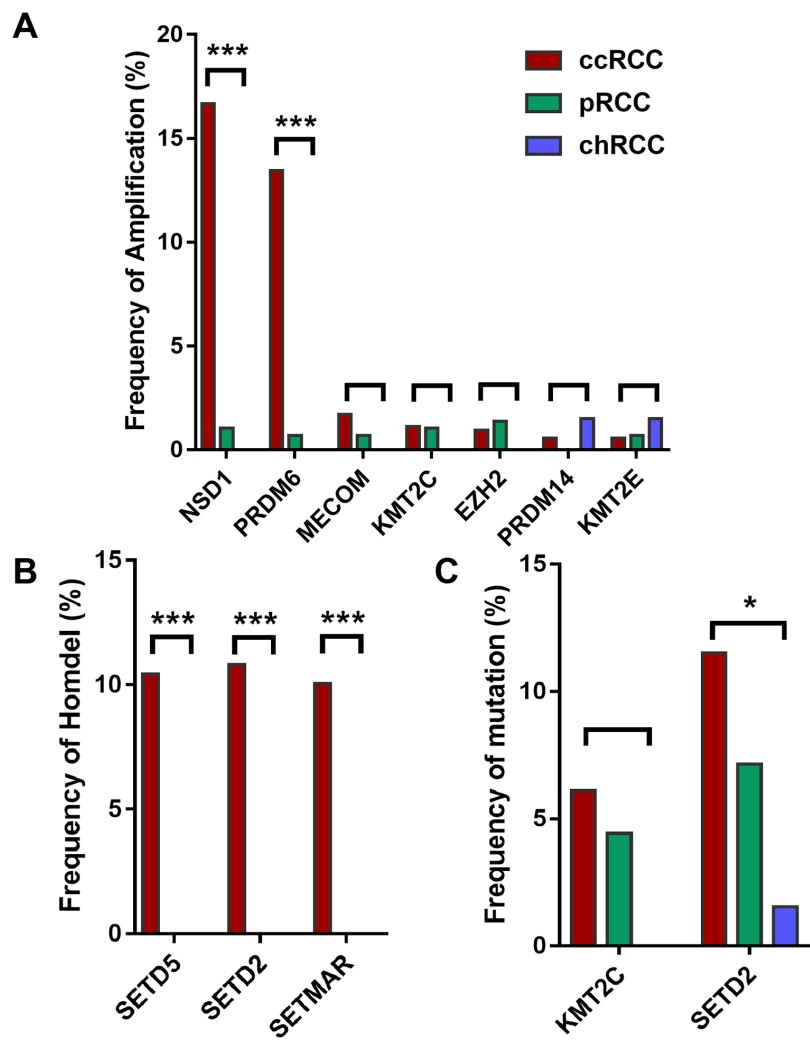

**Figure 1 Frequencies of CNA of HMTs.** (A) High-level amplification of seven HMTs in 528 ccRCC, 288 pRCC, and 66 chRCC samples, $P < 0.001$ in NSD1, $P < 0.001$ in RDM6, $P > 0.05$ in other HMTs; (B) homozygous deletion of two HMTs in 528 ccRCC, 288 pRCC, and 66 chRCC samples, $P < 0.001$ in SETD5, SETD2, and SETMAR; (C) mutation of two HMTs in 213 ccRCC, 113 pRCC, and 65 chRCC specimens, $P > 0.05$ in KMT2C and $P = 0.029$ in SETD2. $^*P < 0.05$ $^{***}P < 0.001$.

($r = 0.596$). Meanwhile, to investigate the degree of discretization of the data sets, the standard deviations of mRNA expression between the ccRCC and non-ccRCC subtypes was performed. As shown in the Table 3, PRDM6 in ccRCC group had a higher degree of discretization than that of in non-ccRCC group.

Expression levels of the 46 HMTs was compared in the heatmap. Compared with non-ccRCC, mRNA levels of 10 HMTs (PRDM1, PRDM8, MECOM, PRDM16, SETD7, PRDM5, ASH1L, NSD1, SUV39H2, and SETDB1) were significantly higher ($P < 0.001$) and 10 HMTs (PRDM12, SUV420H2, SETMAR, SETD1A, SETD2, SETD4, PRDM4, SETD8, DOT1L, and SETD1B) were significantly lower ($P < 0.001$) in ccRCC (Fig. 2; Table S2).

**Table 3 Associations between CNA and expression, and comparison of mRNA expression between ccRCC and non-ccRCC kidney cancer subtypes.**

| Gene | CNA/mRNA Correlation | | | ccRCC /non-ccRCC Comparison T statistics |
|---|---|---|---|---|
| | Spearman | Kendall | Pearson | |
| WHSC1L1 | 0.689 | 0.596 | 0.669 | −0.09 |
| SETDB1 | 0.580 | 0.509 | 0.597 | −0.02 |
| NSD1 | 0.517 | 0.449 | 0.502 | −0.08 |
| SMYD3 | 0.470 | 0.419 | 0.504 | −0.34 |
| EZH2 | 0.462 | 0.414 | 0.459 | −0.07 |
| SMYD2 | 0.433 | 0.392 | 0.427 | −0.14 |
| SUV39H2 | 0.416 | 0.376 | 0.478 | −0.09 |
| PRDM1 | 0.394 | 0.357 | 0.374 | −0.10 |
| PRDM6 | 0.384 | 0.343 | 0.369 | 0.20 |
| ASH1L | 0.380 | 0.345 | 0.389 | −0.08 |
| SETD1A | 0.273 | 0.240 | 0.234 | 0.02 |
| PRDM11 | 0.242 | 0.215 | 0.179 | −0.07 |
| SMYD4 | 0.189 | 0.177 | 0.158 | −0.07 |
| MECOM | 0.176 | 0.159 | 0.156 | −1.32 |
| PRDM8 | 0.162 | 0.154 | 0.164 | −0.22 |
| PRDM13 | 0.161 | 0.159 | 0.103 | −0.29 |
| SMYD1 | 0.159 | 0.157 | 0.185 | 0.06 |
| PRDM10 | 0.157 | 0.149 | 0.133 | −0.10 |
| PRDM4 | 0.086 | 0.089 | 0.087 | −0.26 |
| SUV420H1 | 0.087 | 0.091 | 0.085 | −0.15 |
| SETD1B | 0.071 | 0.066 | 0.058 | −0.07 |
| EHMT1 | 0.070 | 0.064 | 0.079 | 0.01 |
| SETD8 | 0.068 | 0.063 | 0.058 | −0.02 |
| SETD6 | 0.067 | 0.062 | 0.068 | 0.02 |
| SETMAR | 0.063 | 0.058 | 0.043 | −0.23 |
| EZH1 | 0.061 | 0.057 | 0.052 | −0.02 |
| DOT1L | 0.060 | 0.056 | 0.049 | 0.15 |
| SETD3 | 0.059 | 0.055 | 0.072 | −0.10 |
| PRDM15 | 0.023 | 0.019 | 0.031 | 0.05 |
| WHSC1 | 0.022 | 0.019 | 0.010 | −0.03 |
| SETD5 | 0.021 | 0.018 | 0.008 | 0.00 |
| PRDM7 | 0.020 | 0.018 | 0.015 | −0.11 |
| PRDM14 | 0.018 | 0.016 | 0.023 | −0.31 |
| SETDB2 | 0.016 | 0.014 | 0.021 | −0.08 |
| PRDM5 | 0.011 | 0.009 | 0.003 | −0.58 |
| PRDM12 | 0.010 | 0.009 | 0.001 | −0.32 |
| PRDM9 | 0.009 | 0.009 | 0.002 | −0.32 |
| PRDM16 | 0.008 | 0.007 | 0.048 | −1.50 |
| PRDM2 | 0.008 | 0.007 | 0.020 | −0.14 |

(Continued)

| Gene | CNA/mRNA Correlation | | | ccRCC /non-ccRCC Comparison T statistics |
|------|---------|---------|---------|------|
| | Spearman | Kendall | Pearson | |
| SUV39H1 | 0.007 | 0.006 | 0.010 | −0.03 |
| SETD4 | 0.003 | 0.003 | 0.017 | −0.06 |
| SETD2 | 0.002 | 0.002 | 0.002 | 0.01 |
| SETD7 | 0.001 | 0.001 | 0.001 | −0.07 |
| SMYD5 | 0.001 | 0.000 | 0.029 | −0.06 |
| EHMT2 | 0.000 | 0.000 | 0.022 | −0.14 |

**Notes:**
Genes were ranked based on the Spearman correlation coefficient. Significantly higher expression of HMTs in the ccRCC subtype is highlighted in dark gray, and significantly lower expression is highlighted in little gray. T statistics represents the T statistic, which is equivalent to the number of standard deviations of mRNA expression levels between the ccRCC and non-ccRCC subtypes. A positive value means that the ccRCC subtype has a higher value, and a negative value means that the ccRCC subtype has a lower value than the non-ccRCC subtypes samples.

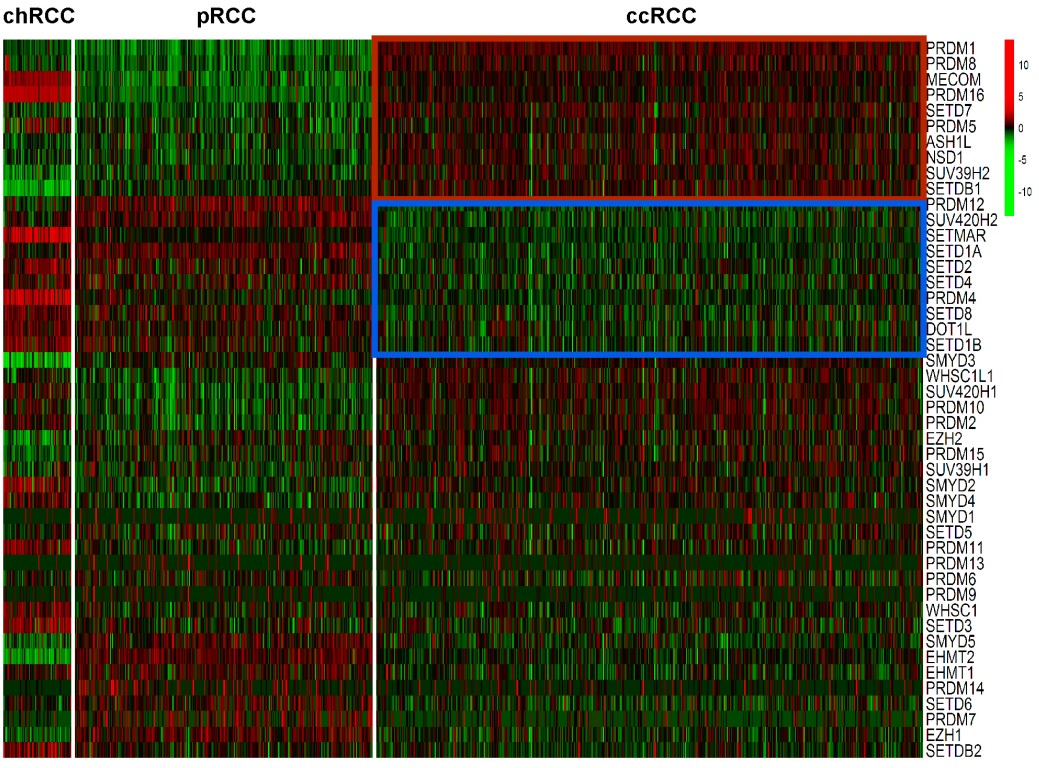

**Figure 2 Heatmap of HMTs expression profiles in different types of renal cell carcinoma.** The kidney cancer samples used in this analysis included 66 chRCC, 288 pRCC, 528 ccRCC kidney cancer samples. Significantly higher-expressed genes (*P* < 0.001) in ccRCC tumors are shown at the top, indicated by a red box; and lower-expressed genes (*P* < 0.001) in ccRCC tumors are indicated by a blue box.

## SETD2 and KMT2C mutations in clear cell renal cell carcinoma

As described previously, we found that SETD2 and KMT2C are most frequently mutated HMTs in ccRCC, at rates of 11.51% and 6.10% (Table 2). A systematic analysis of these mutation profiles was performed in ccRCC samples. Results showed that a total of

**A**

| Mutation Type | SETD2 | KMT2C |
|---|---|---|
| Missense | 21 | 10 |
| Nonsense | 18 | 5 |
| Frameshift Deletion | 8 | 3 |
| Framshift Insertion | 5 | 0 |
| Splice | 6 | 2 |
| Other | 3 | 4 |
| Total | 61 | 24 |

**B**

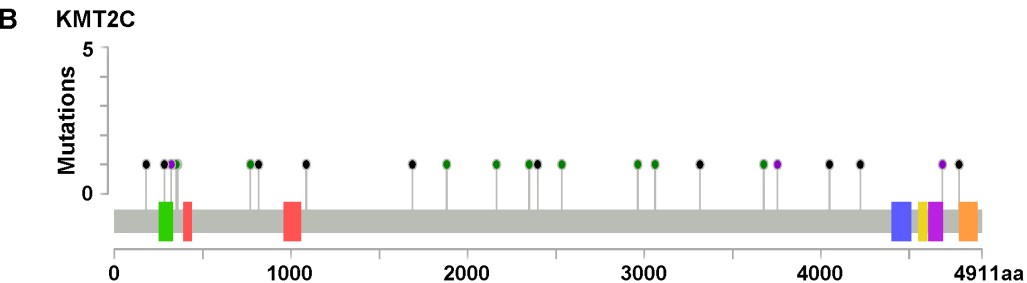

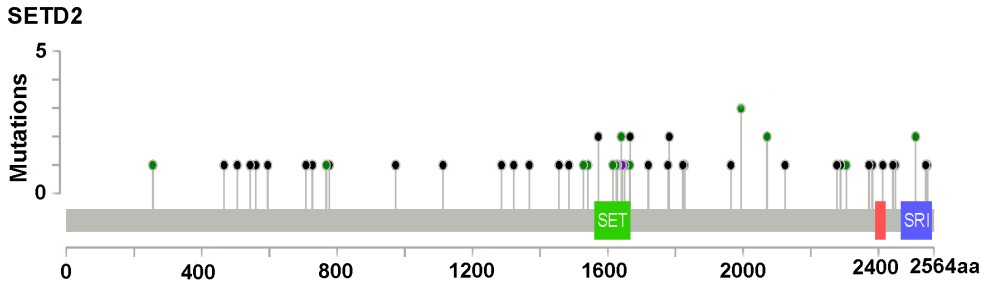

**Figure 3** *KMT2C* and *SETD2* **mutational spectrum in renal cell carcinoma.** (A) Frequency of each mutation type for KMT2C and SETD2 from 213 renal cell carcinoma samples. The data were obtained from The Cancer Genome Atlas database via Cancer Browser. (B) The images show protein domains and the positions of specific mutations of KMT2C and SETD2. A red dot indicates a nonsense mutation, frameshift deletion, insertion, or splice; a green dot indicates a missense mutation; and a black dot indicates an inframe insertion or deletion.

78 SETD2 mutations were found whereas 14 mutations were excluded because their data were untested. A total of 64 mutations were valid, including 18 nonsense mutations, 21 missense mutations, five frameshift insertions, eight frameshift deletions, six splices, and three other mutations. In addition, 24 KMT2C gene mutations were identified, including 10 missense mutations, five nonsense mutations, three frameshift deletions, two splice, and four other mutations (Fig. 3A). A mutation map was performed to display the distribution of SETD2 and KMT2C mutations (Fig. 3B). By systematic analysis of the mutation distribution, we found that SETD2 mutations were more likely to occur at SET domain area. Taking account of the crucial function of SET domain in SETD2, we predicted that mutations at the SET domain might result in the loss of methyltransferases features of SETD2 and poor prognosis of ccRCC patients. Therefore, we performed a

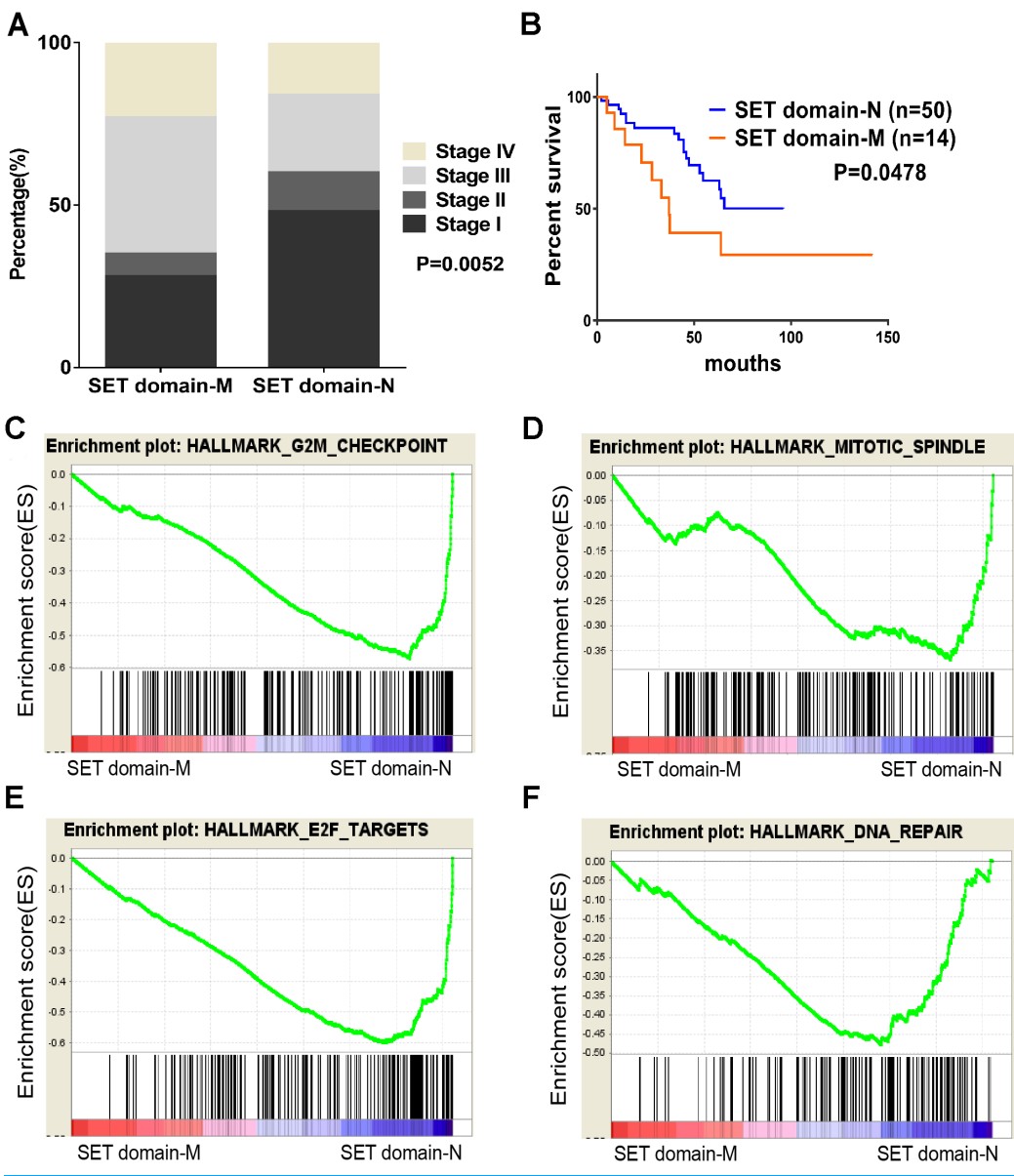

**Figure 4  SET domain mutation VS non-SET domain mutation.** (A, B) Percentage of clinical stage of SETD2 domain mutation vs SETD2 non-SET domain mutation and Kaplan-Meier plots of overall survival associated with SETD2 domain mutation and SETD2 non-SET domain mutation. (C–F) SETD2 SET domain-mutated associated biological signaling pathways. Based on the TCGA dataset, GSEA showed genes associated with cell cycle, mitosis, transcription and DNA repair procedure were signifcantly enriched in SETD2 SET domain-mutated vs SETD2 non-SET domain-mutated tumors.

Kaplan–Meier plots to investigate the clinical features between SETD2 SET domain mutation group and non-SET domain mutation group; results indicated that SETD2 SET domain mutation group was featured with advanced tumor stage and poor prognosis in ccRCC (Fig. 4A).

## Identification of SET-Domain associated biological process by GSEA

To identify SET-Domain associated biological process and function loss of SETD2 on a generalized level, GSEA was performed by using high throughput RNA-sequencing data of the TCGA cohort. The mutation state of SET domain was used as the phenotype label. Among all the predefined Hallmarks gene sets, DNA repair, E2F targets, G2M checkpoint, and mitotic spindle were found to be significantly associated with SET domain mutation in the TCGA cohort (Fig. 4B).

## HMTs CNA and expression and clear cell renal cell carcinoma patient survival

To explore the clinical association of genetic alterations of HMTs in RCC, we investigated the association between CNA, mRNA expression, and overall patient survival in 452 ccRCC samples, 76 ccRCC samples were excluded because their detailed survival data is not available in TCGA database. First of all, samples were divided into the following three groups for each HMT: amp/gain (high-level amplification/low-level gain), diploid, and deletion. For six HMTs (EZH2, NSD1, PRDM6, SETD2, SETD5, and SETMAR), copy number amp/gain and deletion were significantly related to poorer survival in RCC patients ($P < 0.05$). Deletions of KMT2C and PRDM6 were related to shorter survival, however, only amp/gain of EZH2 and PRDM14 was more likely related to poorer survival. More importantly, deletion of NSD1 was significantly related to poorer survival; amp/gain of NSD1 was significantly related to longer survival, compared with diploid or deletion groups (Fig. 5A; Fig. S1).

To analyze the relationship between HMTs expression and overall survival of ccRCC patients, they were divided into low ($n = 226$) and high ($n = 226$) expression groups based on the mRNA expression of each HMT. High mRNA levels of EZH2, PRDM6, SETD5, and SETMAR were significantly associated with shorter survival in ccRCC patients, whereas only high NSD1 expression was correlated with longer survival in ccRCC ($P < 0.05$) (Fig. 5A; Fig. S1).

A multivariate analysis by Cox model ($n = 428$) was performed to investigate the capability to predict poor prognosis of each HMT compared with standard prognostic markers, such as age at diagnosis, gender and tumor stage (stage I–stage IV). Results indicated that amp/gain of ASH1L had a hazard radio (HR), a ratio of death probabilities, of 1.533 compared with non-amp/gain of ASH1L in RCC patients. In addition, deletion of PRDM8 was significantly associated with shorter survival (HR = 1.516, $P < 0.05$) in RCC patients. High mRNA levels of SETD1A (HR = 1.275) and PRDM9 (HR = 1.258) was significantly associated with shorter survival in RCC patients ($P < 0.05$). However, higher expression of PRDM8 (HR = 0.739), EHMT1 (HR = 0.723), ASH1L (HR = 0.711), WHSC1L1 (HR = 0.674), and SUV420H1 (HR = 0.616) was negatively associated with shorter survival in RCC patients ($P < 0.05$). Also, amp/gain of PRDM6 (HR = 0.703), PRDM9 (HR = 0.662), PRDM7 (HR = 0.640), SETD1A (HR = 0.638), NSD1 (HR = 0.598), and DOT1L (HR = 0.592) was negative correlated with shorter survival in RCC patients ($P < 0.05$). (Fig. 5B; Table S3).
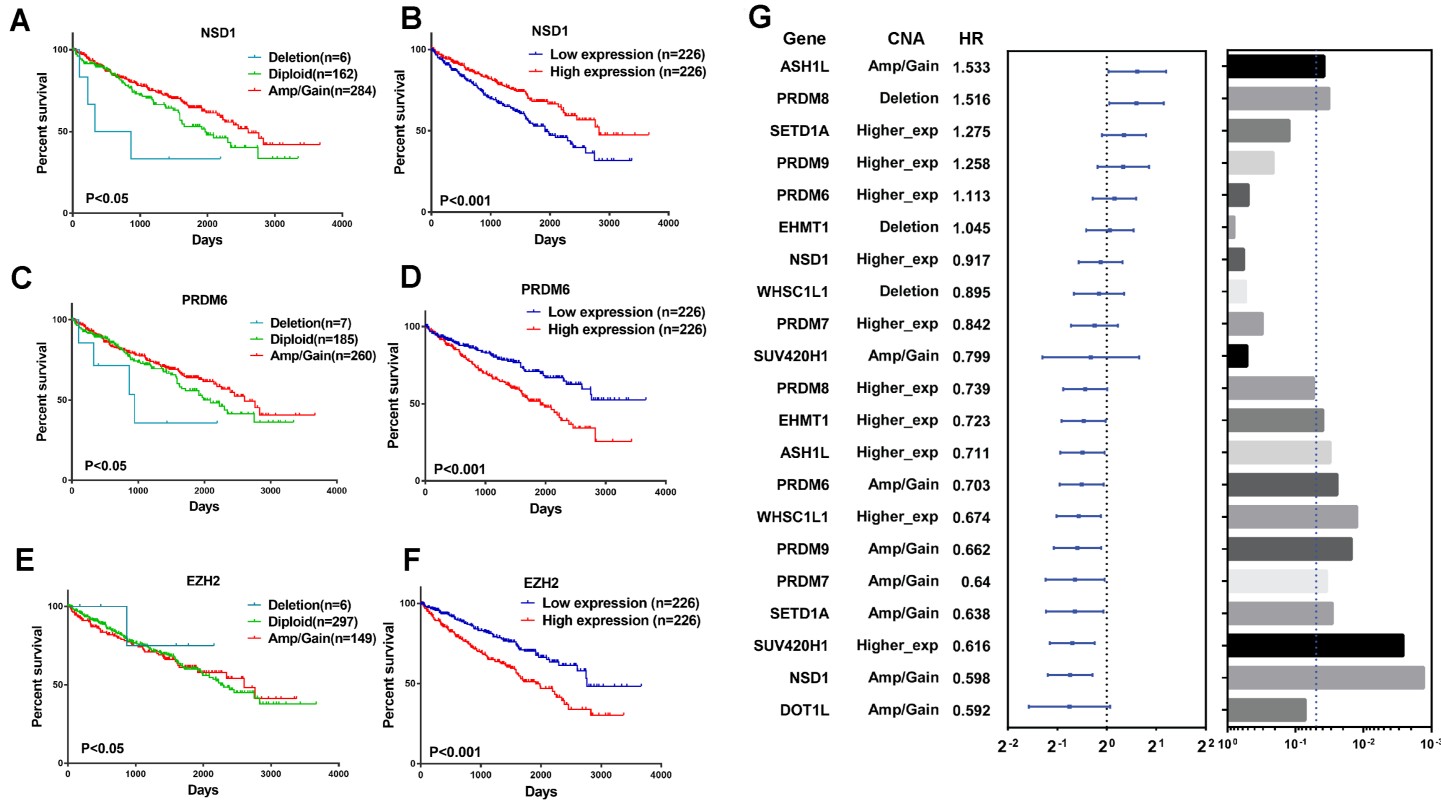

**Figure 5 Clinical outcome with CNA or gene expression level of HMTs.** (A–F) Kaplan-Meier plots of overall survival associated with copy number and mRNA expression levels of three HMTs (NSD1, PRDM6, and EZH2) in renal cell carcinoma. (G) Multivariate analysis (Cox model, n=428) of genetic alteration and gene expression of 11 HMTs (ASH1L, PRDM8, SETD1A, PRDM9, PRDM6, EHMT1, NSD1, WHSC1L1, PRDM7, SUV420H1 and DOT1L) and their HR values, respectively.

## RT-PCR analysis of mRNA expression of important HMTs

Quantitative RT-PCR was performed to measure the expression level of eight important HMTs (NSD1, PRDM6, MECOM, EZH2, ASH1L, SETD1A, WHSC1L1, and SETD2). HK-2, a renal tubular epithelial cell line, was used as the control group. Relative expression of the eight HMTs in kidney cancer cell lines compared with HK-2 cell line was shown in Fig. 6. It showed that mRNA levels of EZH2 were more than fourfold higher in kidney cancer cell lines. In contrast, mRNA levels of WHSC1L1, ASH1L, and NSD1 were more than onefold lower in kidney cancer cell lines. Notably, for SETD2 and MECOM, mRNA levels were more than threefold and sixfold lower in kidney cancer cell lines, respectively. However, mRNA level of PRDM6 did not show significant difference between RCC and control cell lines. These results indicated that there was a correlated change between CNAs and mRNA expression.

## Comprehensive identification of important HMTs in RCC

According to previous data of these CNA, mutation, mRNA expression, and clinical outcome, Table 4 listed the integrative score of important HMTs. Every category counted as "+" when an HMT met the criteria. Especially, SETD2 and KMT2C counted as "++" in CNA/Mutations category because they met the two criteria. As shown in

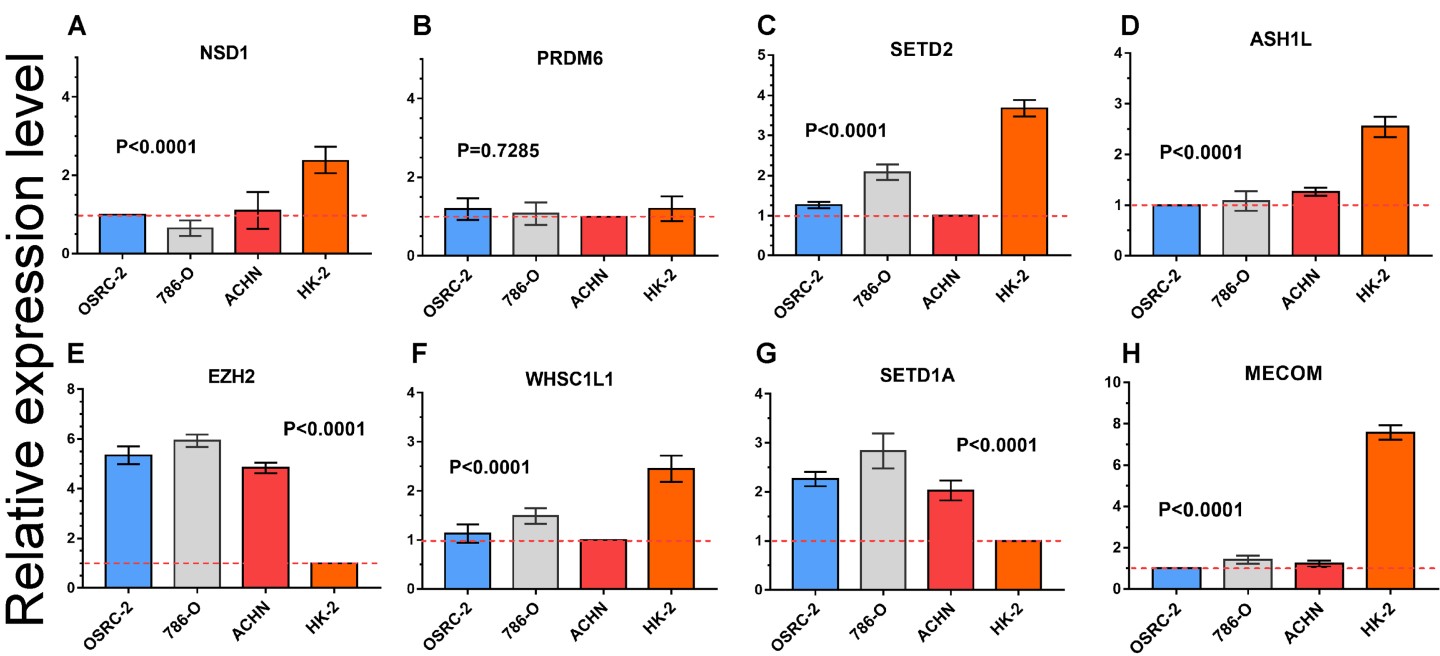

**Figure 6  mRNA expression levels of eight HMTs in three RCC cell lines (OSRC-2, 786-O, and ACHN) and renal tubular epithelial cell line (HK-2) measured by qRT-PCR.** (A–H) mRNA relative expression level of 8 HMTs (NSD1, PRDM6, EZH2, WHSC1L1, SETD2, ASH1L, SETD1A, and MECOM) in OSRC-2, ACHN, 786-O, and HK-2 cell lines, the lowest gene expression level was set as "1" among the four cell lines. Relative expression levels are shown as fold changes compared with "1".           

**Table 4  Integrative identification of critical HMTs in kidney cancer.**

| Gene | CNA/Mutations | CNA/mRNA correlation | Expression | mRNA/survial | CNA/survival | Score |
|---|---|---|---|---|---|---|
| ASH1L | + | | + | + | | 3 |
| PRDM8 | + | | | | | 1 |
| SETD1A | + | | + | | | 2 |
| PRDM9 | + | | | | | 1 |
| PRDM6 | + | | | + | + | 3 |
| EHMT1 | + | | | | | 1 |
| NSD1 | + | + | + | + | + | 5 |
| WHSC1L1 | + | + | + | | | 3 |
| PRDM7 | + | | | | | 1 |
| SUV420H1 | + | | | + | | 2 |
| DOT1L | + | | | | | 1 |
| SETD2 | ++ | | + | + | | 4 |
| EZH2 | + | | + | + | | 3 |
| MECOM | + | | + | | | 2 |
| SETD5 | + | | | + | | 2 |
| SETMAR | + | | | + | | 2 |
| KMT2C | ++ | | | | | 2 |
| KMT2E | + | | | | | 1 |

**Notes:**
CNA/Mutation, amplifcation, deletion or mutation; CNA/mRNA correlation, associations between CNA and gene expression; Expression, altered expression in RCC; mRNA/Survival, mRNA associated with patient survival; CNA/Survival, CNA associated with patient survival.

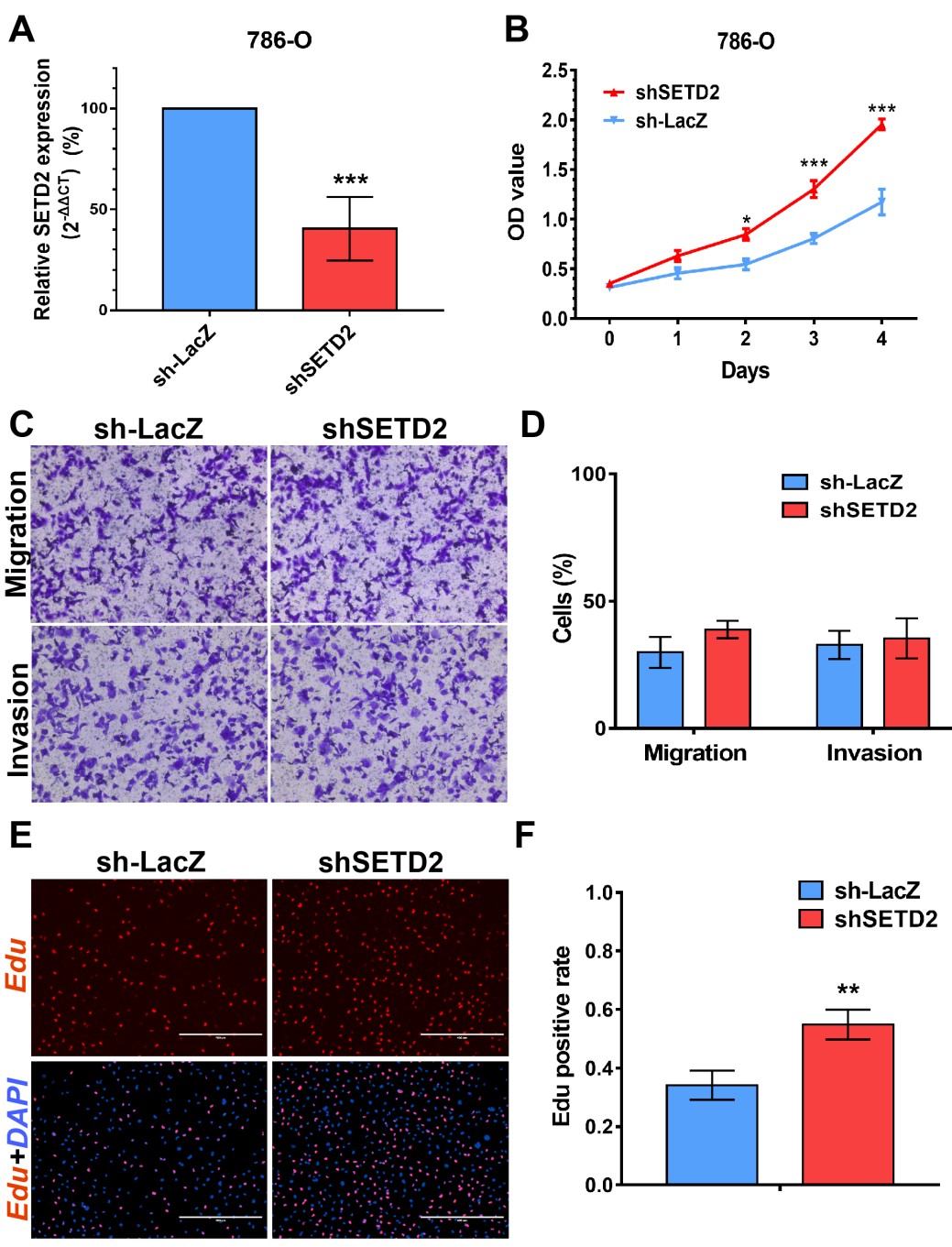

**Figure 7 SETD2 inhibits cell proliferation *in vitro*.** (A) qPCR analysis of relative expression of SETD2 in sh-LacZ and sh—SETD2 cells. (B) MTS assays for 786-O cells without or with stably SETD2 knockdown. (C) and (D) Migration and invasion analyses of Transwell for 786-O cells without or with stably SETD2 knockdown. (E) and (F) EdU assays for 786-O cells without or with stably SETD2 knockdown. $^{*}P < 0.05$ $^{**}P < 0.01$ $^{***}P < 0.001$.

Table 4, six HMTs (ASH1L, PRDM6, NSD1, EZH2, WHSC1L1, SETD2) had a score of more than 2, suggesting that these six HMTs may play important roles in RCC oncogenesis.

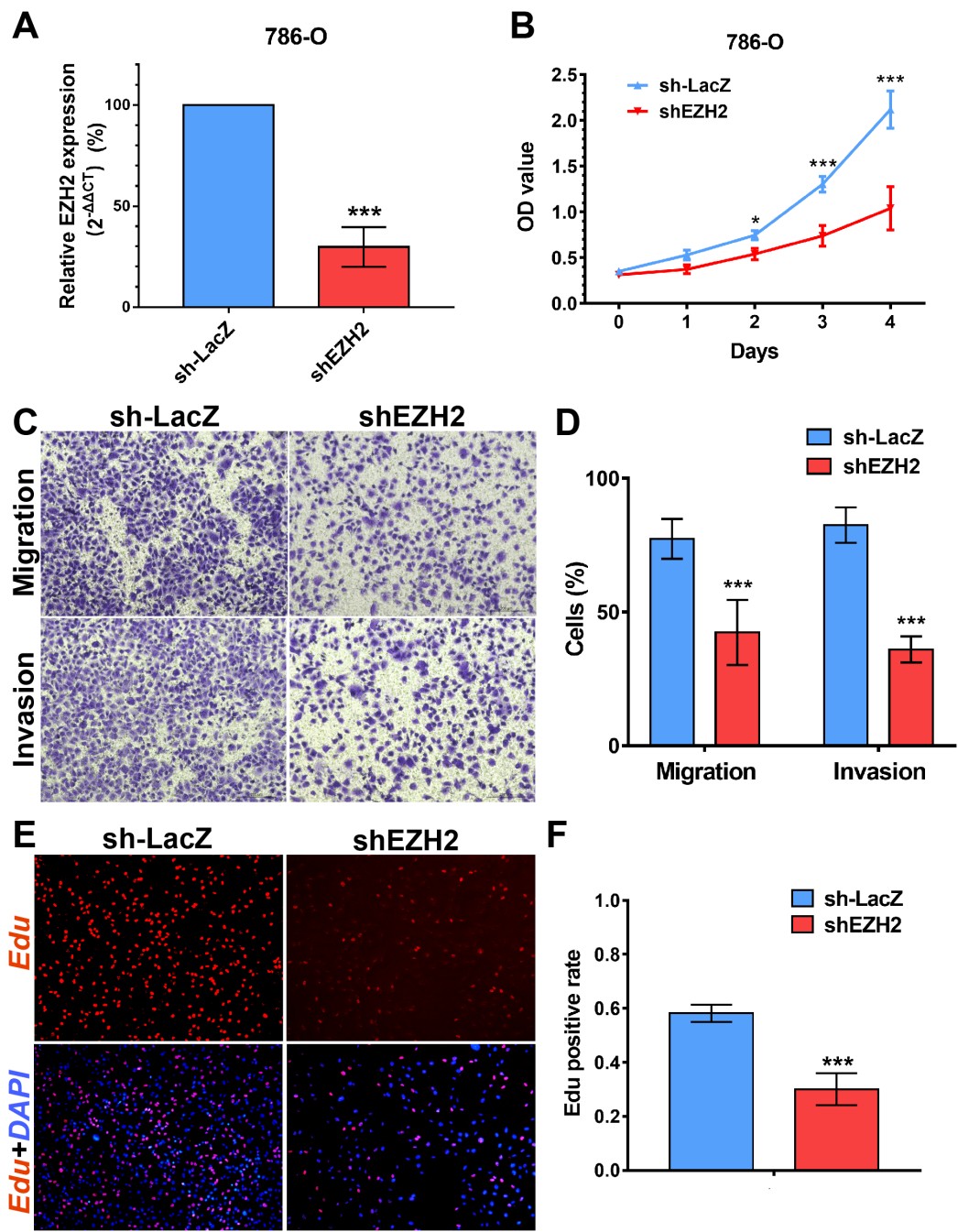

**Figure 8 EZH2 promotes cell proliferation, migration and invasion *in vitro*.** (A) qPCR analysis of relative expression of EZH2 in sh-LacZ and sh—SETD2 cells. (B) MTS assays for 786-O cells without or with stably EZH2 knockdown. (C) and (D) Migration and invasion analyses of Transwell for 786-O cells without or with stably EZH2 knockdown. (E) and (F) EdU assays for 786-O cells without or with stably EZH2 knockdown. $*P < 0.05$ $**P < 0.01$ $***P < 0.001$.

## SETD2 inhibits cell proliferation in vitro, EZH2 promotes cell proliferation, migration and invasion in vitro

We performed MTS assays, Transwell migration/invasion assays and EdU assays in 786-O cell line to detect the function on cell proliferation and migration of SETD2

and EZH2. Assays showed a higher proliferative ability in 786-O cell line with SETD2 knockdown. However, the migration and invasion ability showed no significant differences with or without SETD2 knockdown. Meanwhile EZH2 knockdown reduced the ability of cell proliferation, migration, and invasion significantly. Results mentioned above (Figs. 7 and 8) were consistent with our preceding findings.

## DISCUSSION

We performed a systematic analysis of 50 human HMTs in RCC samples of TCGA and clinical data. Our main findings were listed as follows: (1) A total of 12 HMTs were identified with the highest frequency of genetic alterations; (2) the highest frequency of the top two HMTs with high-level amplification were located in chromosome 5q; whereas the highest frequency of the top three HMTs with homozygous deletion were located in chromosome 3p; (3) correlation between gene expression and CNA was identified: all 46 HMTs showed a positive correlation and three HMTs (NSD1, WHSC1L1, SETDB1) had a Spearman correlation coefficient ($r$) greater than 0.5. WHSC1L1 exhibited the highest correlation; (4) Compared with non-ccRCC, mRNA levels of 10 HMTs (PRDM1, PRDM8, MECOM, PRDM16, SETD7, PRDM5, ASH1L, NSD1, SUV39H2, and SETDB1) were significantly higher and 10 HMTs (PRDM12, SUV420H2, SETMAR, SETD1A, SETD2, SETD4, PRDM4, SETD8, DOT1L, and SETD1B) were significantly lower in ccRCC; (5) the top two mutated HMTs, KMT2C, and SETD2, showed various of mutations at the SET domain in which many of SETD2 mutation were located, leading to loss of their methyltransferase functions; (6) we identified six HMTs (EZH2, NSD1, PRDM6, SETD2, SETD5, and SETMAR) which DNA copy number or mRNA expression level was significantly related to poorer survival in RCC patients; (7) mRNA levels of NSD1, WHSC1L1, ASH1L, SETD2, and MECOM were lower in renal cancer cells compared with renal tubular epithelial cell line whereas EZH2 and SETD1A exhibited the opposite; (8) SETD2 inhibits cell proliferation in vitro, EZH2 promotes cell proliferation, migration, and invasion in vitro.

Oncogenic alterations of HMTs, including amplification, homozygous deletion, and mutation, were associated with various human cancers, including RCC (Qu et al., 2016; Li et al., 2016b; Piva et al., 2015b). EZH2 is overexpressed and mutated frequently in RCC and other types of tumors, contributing to tumorigenic potential of cancer (Chen et al., 2017). SETD2 is also a common tumor suppressor gene at chromosome 3p21, which is found to be mutated frequently in RCC (Piva et al., 2015b; Wang et al., 2015; Ho et al., 2016).

Several studies indicated that aberrations of chromosome 3p deletion and gain of chromosome 5q were uncovered in ccRCC patients and presented their prognostic and diagnostic potential (Nagao et al., 2005; Kluzek et al., 2017; Togo et al., 2016). Taking chromosome 3p as an illustration, ccRCC was characterized by a high frequency of allelic deletion or loss of heterozygosity on chromosome 3p, causing biallelic mutation or promoter hypermethylation of von Hippel-Lindau (VHL) gene. Similarly, our results indicated that SETD2, SETD5, and SETMAR were frequently deleted in ccRCC and they

were located at 3p, implying their potential roles as tumor suppressors. Several researches explored their relationship between methylation of VHL promoter, SETD2 mutation and CNA of other genes located at 3p, including SETD5 and BAP1. As shown in Table 3, WHSC1L1 showed the highest correlation between CNA and expression level and oncogenic potential in ccRCC, which was coincidently consistent with that of in breast cancer (*Liu et al., 2015*). Intriguingly, ccRCC patients were divided into two groups by WHSC1L1 mRNA expression level and the low expression group exhibited poorer prognosis, which was opposite from the role of WHSC1L1 in other cancers (*Saloura et al., 2017*; *Irish et al., 2016*).

SETD2 was most mutated in ccRCC with highest frequency rate of 11.51%. SETD2 SET domain played a critical effect on affecting tumor stage of ccRCC patients. Clinical stage of patients with SETD2 mutations occurred in SET domain region were often higher than those of patients whose SETD2 mutations did not located in SET domain. Taking our results of the GO enrichment analysis into account, among all the predefined Hallmarks gene sets, DNA repair, E2F targets, G2M checkpoint, and mitotic spindle were found to be significantly associated with SET domain mutation, suggesting that SET domain mutation may be involved in ccRCC development and progression through the above cancer-associated biological processes (*Piva et al., 2015a*; *Hacker et al., 2016*; *Park et al., 2016*). The above potential pathway involved in DNA repair, cell circle, dual chromatin, and cytoskeletal remodeling might constitute the reason for poor prognosis in patients with SETD2 mutations. Cell experiments revealed that SETD2 inhibits cell proliferation; however, the migration and invasion ability showed no significant differences. In numerous experimental settings, it has been established that SETD2 regulate cell growth in human cancers (*Jiang et al., 2018*; *Feodorova et al., 2018*). Targeting SETD2 is believed to be a promising strategy for cancer therapy. However, an incomplete understanding of the molecular mechanisms that regulate SETD2 remains a barrier.

Furthermore, the other important HMTs including NSD1, PRDM6, EZH2, SETD5, WHSC1L1, and SETMAR were significantly associated with shorter survival in RCC patients. Except for EZH2, few of them were frequently identified and studied in kidney cancer. SETD5 was reported and identified as a new diagnostic marker in prostate cancer (*Sowalsky et al., 2015*), which was consistent with our research. EZH2 is highly expressed in numerous cancers, including RCC (*Wagener et al., 2010*, *2008*; *Azizmohammadi et al., 2017*; *Lv et al., 2015*). Our findings showed that high EZH2 expression level was associated with tumor cell lines and poorer prognosis and in vitro experiments testified that EZH2 promoted cell proliferation, migration, and invasion in ccRCC. These results were consistent with preceding literature (*Avissar-Whiting et al., 2011*; *Hinz et al., 2009*). Our findings suggest that EZH2 may contribute to renal cancer and raise the possibility that it may be essential for the maintenance of malignant phenotypes of renal cancers.

## CONCLUSIONS

In summary, our findings strongly evidenced that genetic alteration of HMTs may play an important role in generation and development of RCC, which lays a solid foundation for the mechanism for further research in the future.

## ACKNOWLEDGEMENTS

We thank Dr. Ke Chen for his advice.

### Funding

This work was supported by the National Natural Science Foundation of China (81470935, 81370805, 81670645, 81602236), the Chenguang Program of Wuhan Science and Technology Bureau (2015070404010199, 2015071704021644), and the National High Technology Research and Development Program 863 (2014AA020607). The funders had no role in study design, data collection and analysis, decision to publish, or preparation of the manuscript.

### Grant Disclosures

The following grant information was disclosed by the authors:
National Natural Science Foundation of China: 81470935, 81370805, 81670645, 81602236.
Chenguang Program of Wuhan Science and Technology Bureau: 2015070404010199, 2015071704021644.
National High Technology Research and Development Program 863: 2014AA020607.

### Competing Interests

The authors declare that they have no competing interests.

### Author Contributions

- Libin Yan conceived and designed the experiments, performed the experiments, analyzed the data, contributed reagents/materials/analysis tools, prepared figures and/or tables, authored or reviewed drafts of the paper, approved the final draft.
- Yangjun Zhang performed the experiments, analyzed the data, prepared figures and/or tables.
- Beichen Ding performed the experiments, contributed reagents/materials/analysis tools.
- Hui Zhou performed the experiments.
- Weimin Yao performed the experiments.
- Hua Xu conceived and designed the experiments, prepared figures and/or tables, authored or reviewed drafts of the paper, approved the final draft.

### Data Availability

cohort: TCGA Kidney Clear Cell Carcinoma (KIRC).
https://xenabrowser.net/datapages/?cohort=TCGA%20Kidney%20Clear%20Cell%20Carcinoma%20(KIRC)&removeHub=https%3A%2F%2Fxena.treehouse.gi.ucsc.edu%3A443%EF%BC%9B.
cohort: TCGA Kidney Papillary Cell Carcinoma (KIRP).
https://xenabrowser.net/datapages/?cohort=TCGA%20Kidney%20Papillary%20Cell%20Carcinoma%20(KIRP)&removeHub=https%3A%2F%2Fxena.treehouse.gi.ucsc.edu%3A443.

cohort: TCGA Kidney Chromophobe (KICH).
https://xenabrowser.net/datapages/?cohort=TCGA%20Kidney%20Chromophobe%20(KICH)&removeHub=httpss%3A%2F%2Fxena.treehouse.gi.ucsc.edu%3A443.

## Supplemental Information

Supplemental information for this article can be found online at http://dx.doi.org/10.7717/peerj.6396#supplemental-information.

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
