# Peer review of "Genetic alteration of histone lysine methyltransferases and their significance in renal cell carcinoma"

_PeerJ, doi:10.7717/peerj.6396_

## Round 0.1 · original submission · Major Revisions

Take in mind that I will only reconsider the work if you will be able to address all criticisms raised by the referees, including the in vitro experiments they suggested. Please let me know if you need more than 55 days to submit your revision,

Reviewer 1 ·

Basic reporting

1. The authors provided an adequate introduction, but the discussion could be improved by adding more literature on the HMTs with genetic alterations.
2. English need to be improved to remove ambiguous meanings. For instance, in the sentences starting in row 50, 56, 113, 124.
3. Several sentences reporting the results miss the reference to a specific figure (i.e. row 122, 127, 143) so it is difficult to follow the manuscript.
4. The figure 6, relative to the MTS assay, in the paragraph starting in row 219 is missing.
5. In row 122 the author mentioned SETMAR gene that is absent from figure 1b.
6. In several figure panels, such as all panels in fig.6, the Pvalue is missing (also NEs in GSEA analysis).
7. In Fig2 a clustering analysis would be more informative.
8. In fig.2: are MECOM and PRDM16 rally differentially expressed between ccRCC and non-ccRCC?
9. It is not evident why the data in Fig.6 are consistent with the computational analysis. Please explain.
10. In fig.4 the interpretation of panel A is not clear.

Experimental design

The rational of the study is well defined.
The methods should contain more details of the performed analysis. It should be clearer in both methods and figure/table legends which and how many samples in each group are used in all the analysis.

Validity of the findings

The authors should control their statistical findings for the different sample size of the cohorts.

Additional comments

1) The authors integrated TGCA data (mutations, copy number variation, expression level and clinical outcome) of 50 HMT in renal cell carcinoma. The authors indicated, in several sentences, that their integrated analysis suggest a role of some HMTs in renal cell carcinoma development and progression. However, their analysis could suggest a possible general role in renal cell carcinoma biology. To support their claim, they could assess tumor heterogeneity in term of clonal evolution combining all molecular evidences (i.e. somatic mutations, CNV and expression).
2) The manuscript needs a lot of improvement in term of details (methods, figures, figure legends) that are missing; an entire figure on MTS assay is missing although the results are reported in the text.
3) The authors didn’t show properly the “significance” of HMTs in RCC as claimed in the title. To start understanding the functional role, they could perform knockdown of some of the 12 (or 11?) HMTs and perform some cellular assays.

Reviewer 2 ·

Basic reporting

The fonts in the graphs is very variable. It has to be uniform and legible. In some of the later graphs (figure 6 and others), the font is not legible.

Experimental design

The authors have analyzed the published TCGA with a focus on HMT's. But they have made very little effort to prove the validity of their findings experimentally. This weakens the manuscript. The authors analyzed the data from TCGA downloading the data from CBIOPORTAL. The authors did not make an effort to correlate their findings performing invitro experiments other than the mRNA expression analysis.

The p values/statistical significance values should be incorporated in the figures as well to make sure if the analysis is significant.

Validity of the findings

Validity in findings is the weakest. The analysis of gains/amplifications/deletions is reported very nicely. The GSEA analysis including the expression in patient samples is very detailed. The authors mention to have performed migration/invasion assays and EdU assay and mention the results as shown in Figure 6, while Figure 6 is showing mRNA expression data. I think it is better for the manuscript to have significance if the results are incorporated in the figures appropriately.

Additional comments

The authors made a detailed computational analysis of the various HMT's that are deregulated in Renal Cell carcinoma (RCC).

The authors utilized the data that could be downloaded from CBIOPORTAL and the patient sample in TCGA data set is significant.

The authors need to strengthen their analysis with invitro analysis including the contribution of the major HMT's identified including SETD2. Minimal experiments which the authors mentioned in the text such as results obtained from invasion and EdU assays should be incorporated in the manuscript.

The RNA expression analysis could be expanded to few additional cell lines to strengthen their findings or refer to previous studies if published.

The GSEA analysis revealed pathways involved in proliferation. Are there any common genes that are correlating between the GO/GSEA/pathway analysis. If so they should be tested in the cell lines utilized for gene expression of HMT's.

figure 6 was wrongly indicated in the text, please correct it.


In the survival analysis, Please indicate the n for each condition,

---

## Round 0.2 · Minor Revisions

Put particular attention to the suggestions of reviewer number one and make efforts to answer all criticisms they still raised.

Reviewer 1 ·

Basic reporting

English can still be improved.

References are ok.

Figures could be improved.

Experimental design

Research question is well define.

it is not clear if the experiments have been reproduced multiple times.

Methods are described with sufficient details.

Validity of the findings

Most of the statistics are now present in the manuscript.

Additional comments

The authors improved the manuscript in term of discussion, details, English, statistics and methods. They also added the figures relative to some functional analysis (proliferation, migration, invasion) on SETD2 and EZH2.
Since they did not analyze the function of the other “critical” HMTs, the title and the abstract should be more specific on these two genes rather than highlighting a broader analysis. Alternatively, the authors should knock down more “critical” HMTs.
(In table 4, it is indicated how “critical” HMTs are selected. However, the word is not the most indicated for the meaning that the authors attribute to it)

There are still sentences reporting the results but missing the reference to a specific figure or table.

The authors added Table 2 reporting the analysis in Fig.2. I still think that a clustering analysis would be informative. This would also help to have more information for the genes that are not in the blue and red boxes.

Figure Legends need to be improved, in particular fig7 and 8.
The fact that the colors, blue and red, in the histograms are swooped (in term right and left position) in the different panels denotes a lack of attention to the details.

Fig.7c and 8c are unclear.

Fig5a is still missing some statistics.

Fig.6. It is not clear to which comparison the P Values refer to.

The authors did not control their statistical findings for the different sample size of the cohorts.

English can still be improved.

Reviewer 2 ·

Basic reporting

The manuscript has been revised as suggested. Literature cited appropriately. Figures were updated mostly as suggested.

Experimental design

This is a helpful study where the authors did analysis of the epigenetic factors and validated their findings

Validity of the findings

The authors did a good job in validating and the conclusions are appropriate

Additional comments

The authors addressed all the concerns raised and i suggest the manuscript for publication pending editors decision.

---

## Round 0.3 · accepted · Accept

In the new version of the manuscript all criticisms have been addressed.

#